# Metaheuristic Algorithms to Optimal Parameters Estimation of a Model of Two-Stage Anaerobic Digestion of Corn Steep Liquor

Olympia Roeva [1,2,*] and Elena Chorukova [1,3]

1  Department of Bioinformatics and Mathematical Modelling, Institute of Biophysics and Biomedical Engineering, Bulgarian Academy of Sciences, Acad. G. Bonchev Str., Bl. 105, 1113 Sofia, Bulgaria
2  Department of Mechatronic Bio/Technological Systems, Institute of Robotics, Bulgarian Academy of Sciences, Acad. G. Bonchev Str., Bl. 2, 1113 Sofia, Bulgaria
3  Department Biotechnology—Bioremediation and Biofuels, The Stephan Angeloff Institute of Microbiology, Bulgarian Academy of Sciences, Acad. G. Bonchev Str., Bl. 26, 1113 Sofia, Bulgaria
*  Correspondence: olympia@biomed.bas.bg

**Abstract:** Anaerobic Digestion (AD) of wastewater for hydrogen production is a promising technology resulting in the generation of value-added products and the reduction of the organic load of wastewater. The Two-Stage Anaerobic Digestion (TSAD) has several advantages over the conventional single-stage process due to the ability to control the acidification phase in the first bioreactor, preventing the overloading and/or the inhibition of the methanogenic population in the second bioreactor. To carry out any process research and process optimization, adequate mathematical models are required. To the best of our knowledge, no mathematical models of TSAD have been published in the literature so far. Therefore, the authors' motivation is to present a high-quality model of the TSAD corn steeping process for the sequential production of $H_2$ and $CH_4$ considered in this paper. Four metaheuristics, namely Genetic Algorithm (GA), Firefly Algorithm (FA), Cuckoo Search Algorithm (CS), and Coyote Optimization Algorithm (COA), have been adapted and implemented for the first time for parameter identification of a new nonlinear mathematical model of TSAD of corn steep liquor proposed here. The superiority of some of the algorithms has been confirmed by a comparison of the observed numerical results, graphical results, and statistical analysis. The simulation results show that the four metaheuristics have achieved similar results in modelling the process dynamics in the first bioreactor. In the case of modelling the second bioreactor, a better description of the process dynamics trend has been obtained by FA, although GA has acquired the lowest value of the objective function.

**Keywords:** mathematical model; parameter identification; metaheuristic algorithm; anaerobic digestion; corn steep liquor

## 1. Introduction

For many years, hydrogen has been considered an alternative to fossil fuels. Scientists have been aiming to develop an environmentally friendly method to produce hydrogen from biomass by optimizing Anaerobic Digestion (AD) systems [1]. In the AD process, microorganisms decompose biomass in the absence of oxygen. Through AD, food and animal wastes are recycled to produce hydrogen gas which can be subsequently converted into biogas (methane) [2,3].

The utilization of wastewater for hydrogen production through AD is promising as it results in the generation of value-added products and reduces the organic load of the wastewater. For example, Ref.[4] investigates the evaluation of hydrogen production with microbial consortia. The significant variables are optimized using a central composite design, resulting in two mathematical models. The resulting environmental benefits can be found in the removal of 50% of the COD, which could be further improved with the recovery of the other metabolites produced, mainly acetic and butyric acids.

The Two-Stage Anaerobic Digestion (TSAD), in which the hydrogen and methane production takes place in two separate bioreactors, has several advantages over the conventional single-stage process [5,6]. The TSAD permits the selection and the enrichment of different bacteria in each anaerobic digester and increases the stability of the whole process by controlling the acidification phase in the first digester and hence preventing the overloading and/or the inhibition of the methanogenic population in the second digester [5,6].

The production of corn starch and starch-derived products results in large amounts of aqueous by-products [7–9]. Corn Steep Liquor (CSL) is a concentrated liquid by-product derived from water and used in the initial stage of the corn wet milling process. Globally, large amounts of CSL are produced daily and discharged into waterways. Despite its high nutritional value and relatively low cost, currently, CSL is not widely used internationally as a feedstock in the fermentation industry. This is primarily because most corn producers sell off their products to dry milling operations, where the corn is processed for various corn-derived products such as corn flour.

CSL has been used recently as a substrate for AD [2]. An experimental study of two-stage anaerobic biodegradation of corn extract has been performed with mesophilic temperatures in both bioreactors. An automatic mode has been implemented using the developed computer system for monitoring and control. The so obtained experimental data are used here to develop a mathematical model.

For many industrially relevant processes, detailed models are not available due to an insufficient understanding of the underlying phenomena. Mathematical models, which naturally could be incomplete and inaccurate to a certain degree, can still be very useful and effective in describing the effects essential for control, optimization, or understanding of the process.

To the best of our knowledge, no mathematical models of TSAD of corn steep liquor have been published in the literature yet.

In this work, a mathematical model of TSAD of corn steep liquor is proposed. Metaheuristic algorithms are considered for model parameter estimation. The main advantage of metaheuristic techniques, which have been proven to be a good alternative to conventional optimization methods, is that they provide a satisfactory solution for a reasonable computational time [10]. Metaheuristics do not depend on the initial conditions of the model. This advantage allows efficient scanning of a large search space to reach a global extremum.

Two categories of metaheuristics are known: single-based and population-based solution methods [11]. In addition, some authors [12,13] proposed eight different groups—biology-based, physic-based, swarm-based, social-based, music-based, chemistry-based, sport-based, and math-based metaheuristic methods.

Evolutionary algorithms such as Genetic Algorithms (GA) [14,15] and Evolution Strategies [16]; Ant Colony Optimization (ACO) [17], Simulated Annealing (SA) [18], Firefly Algorithm (FA) [19], Artificial Bee Colony (ABC) optimization [20], Cuckoo Search Algorithm (CS) [21], etc., are, among many others, some of the examples of classical metaheuristics. Some of the population-based algorithms such as GA, FA and CS could be considered both as biology-based and swarm-based.

Some of the most powerful nature-inspired metaheuristics have been developed and tested for solving different optimization problems. A comprehensive review of metaheuristic algorithms is presented in [22]. Since optimization is a process of making things as efficient as possible, researchers continuously seek new and modified metaheuristic methods that outperform the existing ones. The application of metaheuristic techniques for solving numerical optimization problems is receiving increasing attention.

A new performance assessment method is discussed in [23]. It considers the influence of the control parameters on the metaheuristic algorithms. The method is demonstrated for GA. As a result, a better solution than the best reported so far in the literature is detected. Recently, new performance metrics of metaheuristic optimization methods have been proposed in [24]. The presented results show that the discussed metrics have a good

distinctive performance for different algorithms. The best-found algorithm is Differential Evolution (DE). In [25], an NP-hard problem is solved based on multi-objective Tabu Search (TS), multi-objective variable neighborhood search, and multi-objective particle mass optimization. The presented results show that the performance of multi-objective TS is better than that of other proposed metaheuristic algorithms.

Real-life problems generally exhibit nonlinear constraints and dynamic components. GAs are well-known metaheuristics extremely applicable to problems with such characteristics. They have been well employed in many fields with still growing recognition [26]. GAs are often used instead of traditional optimization methods. Authors of [27] proposed a hybrid where the crossover and mutation operators of GA are integrated with the Teaching–Learning-Based Optimization and Particle Swarm Optimization (PSO) algorithms. The elitist strategy is utilized to boost evolutionary efficiency. A hybrid GA-PSO is proposed in [28] and compared with CS, PSO, GA, and Simulated Annealing. The results of this research indicate that PSO and GA-PSO produce optimal values for all considered objective functions.

So far, their effectiveness and robustness have been demonstrated in the mathematical modelling of fermentation processes. In [29], GA and the Artificial Bee Colony algorithm are applied to cultivation process modelling. A new approach for simultaneous parameter tuning of the metaheuristics is proposed. An optimization of wheat germ fermentation conditions using an artificial neural network combined with GA is discussed in [30]. Based on the optimized scheme, a 117% improvement is achieved compared to that of the control group. Authors in [31] propose GA for optimizing the productivity of the yeast fermentation process. The proposed GA obtains a higher yield production than the conventional open-loop system. A comparison of 8 modifications of GA (simple genetic algorithms and multi-population ones) is presented in [32] for parameter identification of fed-batch cultivation of *S. cerevisiae*. GA with the sequence of mutation, crossover, and selection operators is significantly faster than the other modifications.

Algorithms such as GA can be very useful, but they still have some drawbacks when dealing with multimodal optimization problems [33]. Some results show that FA outperforms GA and is a powerful algorithm to solve even NP-hard problems [19,34,35]. In [36], an improved FA is utilized to solve the multi-depot vehicle routing problem with time windows. The work proves FA feasibility and shows that modified FA outperforms other competing algorithms (GA, ACO, TS, etc.) in terms of results and competence. Parameter estimation of a proposed hyperbolastic type-I diffusion process applying FA is presented in [37]. The low computational cost of FA makes it especially useful for addressing maximum likelihood estimation in diffusion processes. An interesting performance analysis of distance metrics on the exploitation properties and convergence behaviour of FA is presented in [38]. The optimal algorithm tuning based on unique distance metrics shows a new research area for solving large-scale optimization problems. In [39], the enhanced FA is hybridized with the CLT-based K-means algorithm to achieve optimal global convergence. The results show that the hybrid FA-K-means clustering method demonstrates statistically significant superiority compared to other advanced hybrid search variants—GA, DE, PSO, and Invasive Weed Optimization.

The CS algorithm attracts attention due to its simplicity and efficiency over GA because of using the Lévy flights instead of isotropic random walks [40]. Recently proposed CS algorithm [41] for an energy-efficient robotic mixed-model assembly line balancing problem outperforms GA in terms of obtained objective values. In [42], the CS algorithm employs theoretical anomaly generated by a single structure and different field data sets to estimate the model parameters. As a result, a rapid convergence of the objective function and the model parameters is observed. In [43], a technique combining a CS algorithm and a support vector machine is applied in the prognostic staging of oesophageal cancer based on the prognosis index. The proposed algorithm has the highest prediction accuracy compared to six swarm intelligence algorithms, including the ABC algorithm, FA, Gravitational search algorithm, etc., combined with support vector machine learning techniques.

A newly introduced metaheuristic algorithm, Coyote Optimization Algorithm (COA) [44], adopts an interesting technique to achieve a balance between exploration and exploitation [45], as well as the robustness and stability of the algorithm [46]. Binary COA with a hyperbolic transfer function in a wrapper model is applied to a feature selection problem [47]. The algorithm presents low standard deviations when compared to other metaheuristic algorithms and a great convergence curve. An accurate and stable ultrashort-term wind speed prediction method is achieved using chaotic COA [48]. The authors of [49] propose a novel fault diagnosis method in chemical processes based on the Bernoulli shift coyote optimization algorithm. The results demonstrate that the proposed method outperforms other methods in terms of classification accuracy. In [45], COA is proposed for optimal parameter estimation of a proton exchange membrane fuel cell model. The presented results show the superiority of COA over the other compared methods.

The known results indicate that metaheuristic techniques, such as GA, FA, CS, and COA, are particularly relevant nowadays, frequently preferred and efficiently used for many optimization problems, especially for model parameters optimization [29,31,37,42,45,50]. Moreover, they have not been employed in parameter estimation of TSAD of corn steep liquor model until now which is the motivation to adapt and apply them to the considered optimization problem.

The presented study focuses on the mathematical modelling of anaerobic biohydrogen production from corn steep liquor and the following anaerobic processing of the methane production under mesophilic conditions in a two-stage process. Model parameters identification is performed based on GA, FA, CS, and COA metaheuristic algorithms.

The specific contributions and innovations of this study are as follows:

1.  For the first time, a structure of nonlinear differential equations of two-stage anaerobic digestion of corn steep liquor is evaluated based on real experimental data.
2.  A new mathematical model of two-stage anaerobic digestion is developed. To our knowledge, no such models have been published so far.
3.  Four metaheuristic algorithms (GA, FA, CS, and COA) are adopted and successfully applied to identify the parameters of the model proposed here.
4.  The developed mathematical model could be used further for process investigation and optimization based on process monitoring and control.

The rest of the paper is organized in the following order. Section 2 presents the proposed mathematical model of a TSAD process and short descriptions of the applied metaheuristic algorithms, namely GA, FA, CS, and COA. Section 3 presents the numerical results obtained by the model parameters identification. The results of the comparison of the considered metaheuristics are discussed. Conclusions and further investigations are provided in Section 4.

## 2. Materials and Methods

### 2.1. Mathematical Model of the Two-Stage Anaerobic Digestion Process

The two-stage anaerobic digestion process of corn steep liquor for sequential production of $H_2$ and $CH_4$ is carried out in two separate bioreactors [2]. During the first stage, relatively fast-growing acidogenic microorganisms engaged in the production of volatile fatty acids and $H_2$ are cultivated in the hydrogenic bioreactor $BR_1$. Slow-growing acetogenic and methanogenic bacteria are developed during the second stage in the methanogenic bioreactor $BR_2$. The volatile fatty acids are later transformed into $CH_4$ and $CO_2$ in $BR_2$.

The process dynamics in the cascade $BR_1$ and $BR_2$ are presented, using mass balance, by a set of five Ordinary Differential Equations (ODEs) and two algebraic equations, as follows:

$BR_1$:

$$\frac{dS_1}{dt} = -Y_1\mu_1X_1 + D_1(S_{1in} - S_1),\qquad(1)$$

$$\frac{dX_1}{dt} = \mu_1X_1 - D_1X_1,\qquad(2)$$

$$\frac{dAc_1}{dt} = Y_2\mu_1 X_1 - D_1 Ac_1, \tag{3}$$

$$Q_{H_2} = Y_{H_2}\mu_1 X_1, \tag{4}$$

$$\mu_1 = \frac{\mu_{1max} S_1}{K_{S_1} + S_1}. \tag{5}$$

BR$_2$:

$$\frac{dX_2}{dt} = \mu_2 X_2 - D_2 X_2, \tag{6}$$

$$\frac{dAc_2}{dt} = -Y_3\mu_2 X_2 + D_2(Ac_1 - Ac_2), \tag{7}$$

$$Q_{CH_4} = Y_{CH_4}\mu_2 X_2, \tag{8}$$

$$\mu_2 = \frac{\mu_{2max} Ac_2}{K_{S_2} + Ac_2}. \tag{9}$$

The system (1)–(5) describes the dynamics of the substrate concentration ($S_1$), [g/L]; the microbial biomass concentration ($X_1$), [g/L]; the product (acetate) formation ($Ac_1$) [g/L] in BR$_1$. The algebraic equation describes the flow rate of the hydrogen ($Q_{H_2}$) [dm$^3$/L·h] in the gas phase of BR$_1$. $S_{1in}$ [g/L] is the concentration of the input substrate. For the specific growth rate $\mu_1$ [day$^{-1}$] of hydrogen-producing microorganisms, a Monod-type function was adopted. $D_1$ [day$^{-1}$] is the dilution rate for the first bioreactor BR$_1$; $\mu_{1max}$ [day$^{-1}$] and $K_{S_1}$ [g/L] are Monod kinetic coefficients; and $Y_1$, $Y_2$ and $Y_{H_2}$ are yield coefficients, [g/g].

The system (6)–(8) describes a one-step transformation of the inlet acetate $Ac_1$ (coming from BR$_1$) into methane by methanogenic microorganisms. A Monod type function was also adopted for the specific growth rate of the methanogenic biomass. In the model, $X_2$ is the microbial biomass concentration in BR$_2$ [g/L], $Ac_2$ is the acetate concentration in BR$_2$ [g/L], $Q_{CH_4}$ is the methane flow rate [dm$^3$/L·day], $D_2$ [day$^{-1}$] is the dilution rate for the second bioreactor BR$_2$, $\mu_2$ is the specific growth rate (Monod type) of methanogens [day$^{-1}$], $Y_{CH_4}$ and $Y_3$ are yield coefficients, and $\mu_{2max}$ [day$^{-1}$] and $K_{S_2}$ [g/L] are kinetic coefficients.

The so presented model structure is used for the first time for modelling the corn steep liquor TSAD process considered here based on real experimental data.

### 2.2. Metaheuristic Optimization Algorithms

The model parameters identification problem of the TSAD process of corn steep liquor is solved using different metaheuristic algorithms. Four population-based metaheuristics, namely GA, FA, CS, and COA, are adopted and applied to the problem considered here. The chosen algorithms have proved their effectiveness and feasibility in their successful application to a wide range of problems [26,33,45,51,52]. A very brief description of each algorithm is presented below.

#### 2.2.1. Genetic Algorithm

GA originated from the studies on cellular automata of John Holland [14]. Developed as an abstraction of the evolutionary process, GA typically operates on vectors whose elements belong to the binary alphabet {0, 1}. They use a recombination operator with background mutation. A population of a certain number of individuals is retained, where each individual is a promising solution to the problem under consideration. Each individual is measured and its "fitness" is estimated in proportion to the corresponding value of the objective function. The more suitable individuals are listed in a new population. In order to form a new solution [15], some of these individuals are altered using "genetic" operators. As a unary operation, mutation produces a small change in one individual, while higher order operations as crossover combine data from several individuals to create a new one. The algorithm converges in a certain number of iterations. The best individual from the last generation is regarded as a near-optimum (reasonable) solution to the problem.

The pseudocode which describes the functioning of GA is shown in Algorithm 1.

---

**Algorithm 1** Pseudocode of GA

---

1:   **begin**
2:      Define
3:         GA parameters—generation gap, population size $n$, crossover rate, mutation rate, number of generations *MaxGen*
4:         GA operators—selection, crossover, mutation
5:         Set generation number to zero ($t = 0$)
6:         Initialize usually random population of individuals ($P(0)$)
7:         Evaluate fitness of all individuals of the initial population
8:            **while** ($t <$ MaxGeneration) **do**
9:              Increase the generation number
10:             Select a sub-population (select $P(i)$ from $P(i-1)$)
11:             Recombine the genes of selected parents (recombine $P(i)$)
12:             Perturb the mated population stochastically (mutate $P(i)$)
13:             Evaluate the new fitness (evaluate $P(i)$)
14:            **end while**
15:   **end begin**

---

### 2.2.2. Firefly Algorithm

FA was introduced by Xin-She Yang [19] as a new metaheuristic algorithm mimicking the nature of fireflies. Different types of fireflies exhibit unique flashing patterns with two primary functions to interact with others: drawing the attention of potential mating partners and attracting potential victims. The flashing is no less significant as a defence mechanism. Idealizing some of the characteristics of the fireflies' flashing lights allows them to be related to the objective functions of new metaheuristic algorithms which need to be optimized.

According to [53], the main steps of FA can be outlined in Algorithm 2.

---

**Algorithm 2** Pseudocode of FA

---

1:   **begin**
2:      Define
3:         Light absorption coefficient $\gamma$
4:         Initial attractiveness $\beta_0$
5:         Randomization parameter $\alpha$
6:         Objective function $f(x)$, where $x = (x_1, \ldots, x_d)^T$
7:         Generate initial population of fireflies xi ($i = 1, 2, \ldots, n$)
8:         Determine light intensity $I_i$ at $x_i$ via $f(x_i)$
9:         **while** ($t <$ MaxGeneration) **do**
10:         for $i$ = 1: $n$ all n fireflies **do**
11:          **for** $j$ = 1: $i$ all n fireflies **do**
12:           **if** ($I_j > I_i$) **then**
13:            Move firefly $i$ towards $j$
14:           **end if**
15:           Attractiveness varies with distance $r$ via $\exp[-\gamma r^2]$
16:           Evaluate new solutions and update light intensity
17:          **end for** $j$
18:         **end for** $i$
19:         Rank the fireflies and find the current best
20:         **end while**
21:         Postprocess results and visualization
22:   **end begin**

---

### 2.2.3. Cuckoo Search Algorithm

Known for their aggressive reproduction strategy [21], cuckoos not only lay their eggs in other birds' nests, but they may also remove others' eggs with the purpose to secure their generation. This specific cuckoo's brood parasitism reduces the possibility of their eggs

being abandoned and thus increases the cuckoo's reproductivity. The interesting breeding behaviour is implemented in the CS metaheuristic algorithm for resolving optimization problems [21]. The CS algorithm assumes the eggs in the host's nest to be a set of potential solutions to an optimization problem, while the cuckoo egg itself is interpreted as a new one. At each iteration of the algorithm, these new and likely better solutions are used to achieve a very good final solution to the problem.

The pseudocode of the CS algorithm based on [21] is presented in Algorithm 3.

---

**Algorithm 3** Pseudocode of CS algorithm

---

1:   **begin**
2:     **Define**
3:       Number of nests $n$
4:       Switching parameter $p_a$
5:       Objective function $f(x)$, $x = (x_1, \ldots , x_d)^T$
6:       Generate initial population of $n$ host nests $x_i$ $(i = 1, 2, \ldots , n)$
7:         **while** ($t$ < MaxGeneration) or (stop criterion) **do**
8:           Get a cuckoo randomly by Lévy flights
9:           Evaluate its quality or fitness value $f_i$
10:          Choose a nest among $n$ (say, $j$) randomly
11:            **if** ($f_i < f_j$)
12:              Replace $j$ by the new solution $i$
13:            **end if**
14:          A fraction ($p_a$) of worse nests are abandoned
15:          New solutions (nests) are built
16:          Keep the best solutions, i.e. nests with quality solutions
17:          Rank the solutions and find the current best
18:        **end while**
19:        Postprocess results and visualization
20:   **end begin**

---

### 2.2.4. Coyote Optimization Algorithm

COA is a new population-based metaheuristic algorithm for optimization inspired by the *Canis latrans* species that dwells mainly in North America. The algorithm designed by Pierezan and Dos Santos Coelho [44] considers the social organization of the coyotes and their adaptation to the environment. The population of coyotes is divided into several packs $N_p$ with a predefined number $N_c$ of coyotes each. The total population is formed by $N_p \times N_c$. Each coyote is a possible solution to the optimization problem and its social condition is the cost of the objective function. COA provides new mechanisms for balancing exploration and exploitation in the optimization process [44].

The pseudocode of the COA according to [44] is presented in Algorithm 4.

---

**Algorithm 4** Pseudocode of COA algorithm

---

1:   **begin**
2:     Define
3:       $N_p$ packs with $N_c$ coyotes each
4:       probability of eviction of a coyote *p_leave*
5:       Verify the coyote's adaptation
6:         **while** stopping criterion is not achieved **do**
7:           **for** each $p$ pack **do**
8:             Define the alpha coyote of the pack
9:             Compute the social tendency of the pack
10:            **for** each $c$ coyotes of the $p$ pack **do**
11:              Update the social condition
12:              Evaluate the new social condition
13:              Adaptation
14:            **end for**

**Algorithm 4** *Cont.*

| | |
|---|---|
| 15: | Compute $\omega$ and $\phi$. |
| 16: | **if** $\phi = 1$ **then** |
| 17: | The pup survives and the only coyote in $\omega$ dies |
| 18: | **elseif** $\phi > 1$ **then** |
| 19: | The pup survives and the oldest coyote in $\omega$ dies |
| 20: | **else** |
| 21: | The pup dies. |
| 22: | **end if** |
| 23: | **end for** |
| 24: | Transition between packs |
| 25: | Update the coyotes' ages |
| 26: | **end while** |
| 27: | Select the best adapted coyote |
| 28: | **end begin** |

## 3. Results and Discussion

### 3.1. Simulation Setup

The proposed mathematical model consists of a set of five ODEs and two algebraic equations (Equations (1)–(9)) thus represented by seven dependent state variables $x = [X_1\ S_1\ Ac_1\ Q_{H_2}\ X_2\ Ac_2\ Q_{CH_4}]$ and nine unknown parameters divided into two groups: $p_1 = [\mu_{1max}\ K_{S_1}\ Y_1\ Y_2\ Y_{H_2}]$ and $p_2 = [\mu_{2max}\ K_{S_2}\ Y_3\ Y_{CH_4}]$.

Each individual of the population (chromosome in GA, firefly in FA, nest in CS, and coyote in COA) represents a potential solution to the problem—the corresponding unknown model parameter vector $p_1$ or $p_2$. Based on [54,55] and the authors' expertise, each parameter of the model (1)–(9) has been coded in the specific range (lower bound (*Lb*) $\leq$ parameter $\leq$ upper bound (*Ub*)) as follows:

$$
\begin{aligned}
&0.01 \leq \mu_{1max} \leq 0.8;\ 0.1 \leq K_{S_1} \leq 50;\ 0.1 \leq Y_1 \leq 15; \\
&0.001 \leq Y_2 \leq 40;\ 0.01 \leq Y_{H_2} \leq 15; \\
&0.001 \leq \mu_{2max} \leq 0.8;\ 0.0001 \leq K_{S_2} \leq 5;\ 0.1 \leq Y_3 \leq 50;\ 0.1 \leq Y_{CH_4} \leq 10.
\end{aligned} \tag{10}
$$

Each initial solution for all four metaheuristic algorithms has been generated based on

$$
x_j = rand * (Ub - Lb) + Lb. \tag{11}
$$

Further improvement of the solution has been made based on the specific algorithm inspired by the considered behaviour in nature in the case of GA, FA, CS, and COA. New solutions are generated taking into account the bounds (constraints) of the model parameters. In addition, each newly improved solution is tested to see if it is within the defined bounds.

The metaheuristic algorithms have been executed on Intel® Core™i7-8700 CPU @ 3.20 GHz, 3192 MHz (ASBIS Bulgaria Ltd., Sofia, Bulgaria), 32 GB Memory (RAM), Windows 10 pro (64 bit) operating system. Matlab and Simulink R2019a environment are used. Based on the presented pseudocodes (Algorithms 1–4), the algorithms are implemented in Matlab code. Mathematical model of corn steep liquor TSAD process (Equations (1)–(9)) is presented as a Simulink file (see Figure 1). Solver options are fixed-step size of 0.01 and ode4 (Kunge-Kutta) with TIMESPAN = [0 12].

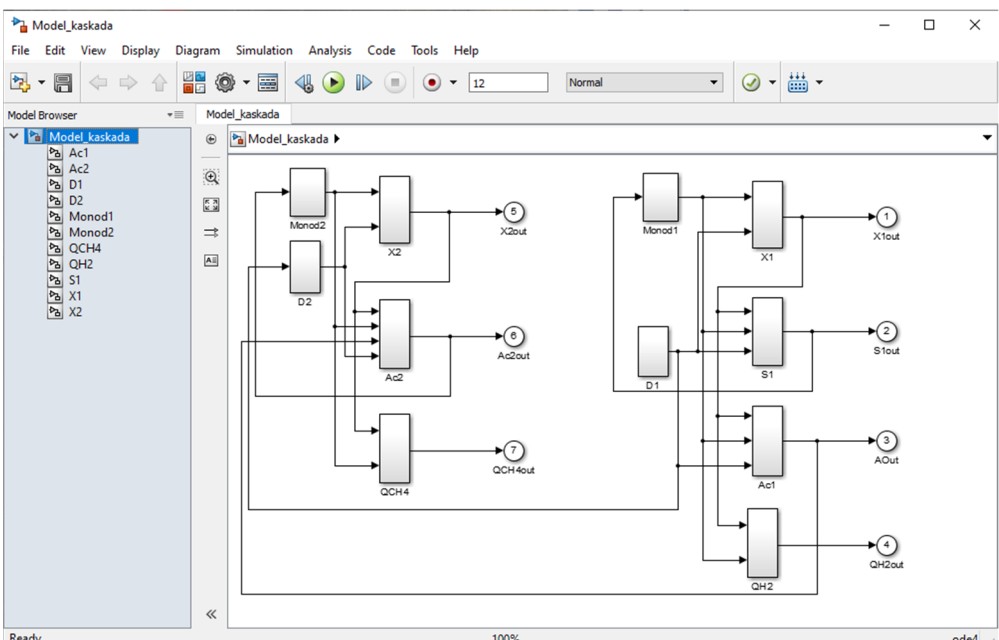

**Figure 1.** Mathematical model of corn steep liquor TSAD process ((1)–(9)) described in Simulink.

To achieve the best metaheuristic algorithm performance, necessary adjustments of the GA, FA, CS and COA parameters depending on the problem domain have been provided [23]. Starting from an already explored range of parameters variation in the case of GA [29,56,57], FA [19,34–36,38], CS [21,42,43,50], and COA [44–46] and taking into account the characteristics of the particular identification problem here, several pre-tests (based on trial-and-error method) have been performed to tune the parameters of GA, FA, CA, and COA. The four algorithms have been run for 100 iterations, which preliminary tests showed was sufficient for the algorithms to converge. The main GA functions are as follows: fitness function—linear ranking, selection function—roulette wheel selection, crossover function—double point crossover, and mutation function—bit inversion. A binary chromosome representation with a precision of binary representation 20 is used.

The set of optimal values used for the main parameters of the algorithms after a series of adjustments is listed in Table 1.

To ensure a fair comparison of the considered stochastic metaheuristics, each algorithm is run 30 times for 100 iterations; the initial solution is generated based on Equation (11) and the model parameter constraints (*Lb* and *Ub*) are as presented in (10).

A two-stage identification scheme is used. In the first step, the model parameters of the system (1)–(5), namely $p_1 = [\mu_{1max}\ K_{S_1}\ Y_1\ Y_2\ Y_{H_2}]$ (search space dimension $D = 5$), have been estimated based on the objective function $J_1$:

$$J_1 = \sum_{i=1}^{n} \left\{ \left[ H_{2exp}(i) - H_{2mod}(i) \right] \right\}^2 \rightarrow min, \tag{12}$$

where $n$ is the length of the data vector for the state variable $H_2$; $H_{2exp}$ is known experimental data; $H_{2mod}$ is a model prediction with a given set of the parameters.

On the second step, the model parameters $p_2 = [\mu_{2max}\ K_{S_2}\ Y_3\ Y_{CH_4}]$ ($D = 4$) have been estimated based on the objective function $J_2$:

$$J_2 = \sum_{i=1}^{m} \left\{ \left[ CH_{4exp}(i) - CH_{4mod}(i) \right] \right\}^2 \rightarrow min, \tag{13}$$

where $m$ is the length of the data vector for the state variable $CH_4$; $CH_{4exp}$ is known experimental data; $CH_{4mod}$ is a model prediction with a given set of the parameters.

**Table 1.** The optimal values of the main parameters of the metaheuristic algorithms.

| GA | Value |
|---|---|
| population size $n$ | 100 |
| generation gap | 0.97 |
| crossover rate | 0.80 |
| mutation rate | 0.05 |

| FA | Value |
|---|---|
| number of fireflies $n$ | 20 |
| initial attractiveness $\beta_0$ | 1 |
| light absorption coefficient $\gamma$ | 1 |
| randomization parameter $\alpha$ | 0.2 |

| CS | Value |
|---|---|
| number of nests $n$ | 20 |
| switching parameter $p_a$ | 0.25 |
| Lévy exponent $\lambda$ | 1.5 |

| COA | Value |
|---|---|
| number of packs $N_p$ | 30 |
| number of coyotes $N_c$ | 100 |
| probability of eviction of a coyote $p\_leave$ | $0.0005 \times N_c^2$ |
| scatter probability $P_s$ | $1/D$ |
| association probability $P_a$ | $(1-P_s)/2$ |

### 3.2. Simulation Results

A series of parameter identification procedures of the proposed model (Equations (1)–(9)) using tuned GA, FA, CS, and COA has been performed. Due to the stochastic nature of the algorithms, a meaningful statistical analysis requires each metaheuristic to be run at least 30 times.

The best model parameter estimations obtained for models (1)–(9) with the corresponding values of the objective functions are summarized in Table 2.

**Table 2.** Model parameter estimations.

| | $\mu_{1max}$ | $K_{S_1}$ | $Y_1$ | $Y_2$ | $Y_{H_2}$ | $J_1$ | $\mu_{2max}$ | $K_{S_2}$ | $Y_3$ | $Y_{CH_4}$ | $J_2$ |
|---|---|---|---|---|---|---|---|---|---|---|---|
| GA | 0.012 | 0.944 | 0.067 | 14.671 | 11.289 | 0.0035 | 0.517 | 0.812 | 0.029 | 0.543 | 0.0761 |
| FA | 0.017 | 1.100 | 0.222 | 10.276 | 8.117 | 0.0035 | 0.443 | 0.919 | 0.100 | 0.989 | 0.1075 |
| CS | 0.010 | 1.139 | 0.010 | 0.122 | 13.727 | 0.0036 | 0.077 | 0.001 | 19.781 | 2.298 | 0.0913 |
| COA | 0.012 | 1.004 | 12.137 | 5.702 | 11.598 | 0.0035 | 0.029 | 0.0001 | 24.272 | 7.933 | 0.0940 |

The observed results are very interesting. Four distinct models are obtained based on the applied metaheuristic algorithms. The parameter estimations presented in Table 2 are generally satisfying and relevant to their physical meaning. The estimates of the maximum specific growth rate of the microbial biomass concentration in BR$_1$ $\mu_{1max}$ as well as the saturation constant $K_{S_1}$ are almost identical for the four algorithms. Yield coefficient $Y_{H_2}$ is also estimated to the same order of magnitude by all algorithms. For BR$_2$, the results are different. Both GA and FA provide similar results, while the estimates of CS and COA, especially for the parameter $K_{S_2}$, differ by order of magnitude. In the case of dynamics in BR$_1$, GA and FA result in similar models, while in the case of process dynamics in BR$_2$, similar models are obtained by CS and COA, except the estimates of parameter $K_{S_2}$.

Since there are no reported parameter values for the proposed model or similar, it is not possible to determine which parameters, i.e., which model is the best. In such cases, a graphical comparison can clearly show if the model follows the experimental data, i.e., the presence or absence of systematic deviations between the model predictions and the experimental measurements. Such a quantitative measure is also an important criterion for

the adequacy of the models proposed here. The model predictions of the state variables $Q_{H_2}$ and $Q_{CH_4}$, based on GA, FA, CS, and COA estimated set of model parameters, have been compared to the experimental data of the TSAD of corn steep liquor in Figure 2. $J$ is the value of the mean least square error for each model (Equations (12) and (13)).

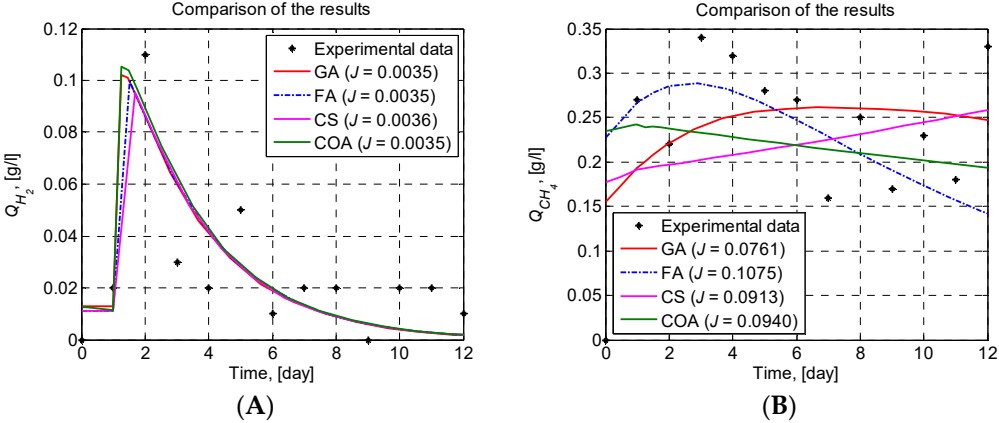

**Figure 2.** Graphical comparison of the results. (**A**) $Q_{H_2}$ dynamics; (**B**) $Q_{CH_4}$ dynamics.

As can be seen in Figure 2, the models' dynamics of the state variable $Q_{H_2}$ in the $BR_1$ are very similar. Therefore, the best model has to be determined based on the results for $BR_2$. In $BR_2$, the dynamics of the state variable $Q_{CH_4}$ predicted by GA, FA, CS, and COA are different. Although the resulting GA model has the lowest $J$ value ($J_2 = 0.0761$), the graphical comparison shows that the resulting FA model is better. The FA model describes the experimental data in the best way, following the trend of the $Q_{CH_4}$ dynamics. The CS and COA models have similar $J$ values, but the resulting models' dynamics do not follow the data trend very well. It should be mentioned that the experimental data are raw, unprocessed data. No filtering is applied in order to test the performance of the metaheuristics for a real nonlinear complex problem.

The performance of the considered metaheuristic algorithms has been statistically evaluated by comparing the observed average value, standard deviation (SD), and the median of the estimated model parameters and the obtained objective function value $J_2$. A statistical analysis of the suggested algorithms in terms of average, SD, and median on the results for the second identification step is presented in Table 3. Only the second set of estimated model parameters, observed in the case of modelling the processes in $BR_2$, is presented and discussed because of their crucial role and significance.

**Table 3.** Statistical analysis of the results.

| Algorithm | | $J_2$ | $Y_3$ | $Y_{CH_4}$ | $\mu_{2max}$ | $K_{S_2}$ |
|---|---|---|---|---|---|---|
| | average | 0.09552 | 0.14501 | 5.32245 | 0.18386 | 0.36572 |
| GA | SD | 0.01732 | 0.26303 | 8.04380 | 0.14778 | 0.41212 |
| | median | 0.08833 | 0.05033 | 1.74911 | 0.14962 | 0.35478 |
| | average | 0.11683 | 0.11048 | 1.10871 | 0.46463 | 0.94210 |
| FA | SD | 0.00936 | 0.01077 | 0.18236 | 0.03767 | 0.03814 |
| | median | 0.11686 | 0.10858 | 0.99504 | 0.44297 | 0.92456 |
| | average | 0.10984 | 0.21904 | 4.26557 | 0.40936 | 0.26250 |
| CS | SD | 0.01361 | 0.53544 | 2.99461 | 0.28370 | 0.32967 |
| | median | 0.10693 | 0.00138 | 3.07739 | 0.45413 | 0.11308 |
| | average | 0.11454 | 0.28767 | 6.18897 | 0.37877 | 0.21275 |
| COA | SD | 0.00741 | 0.23002 | 3.01675 | 0.22794 | 0.21626 |
| | median | 0.11521 | 0.21474 | 6.85870 | 0.37701 | 0.14627 |

Box plot diagrams are commonly used to present summary statistics as a standard technique for analyzing the distribution of the model estimates [58,59]. Box plots of the obtained results are shown in Figure 3.

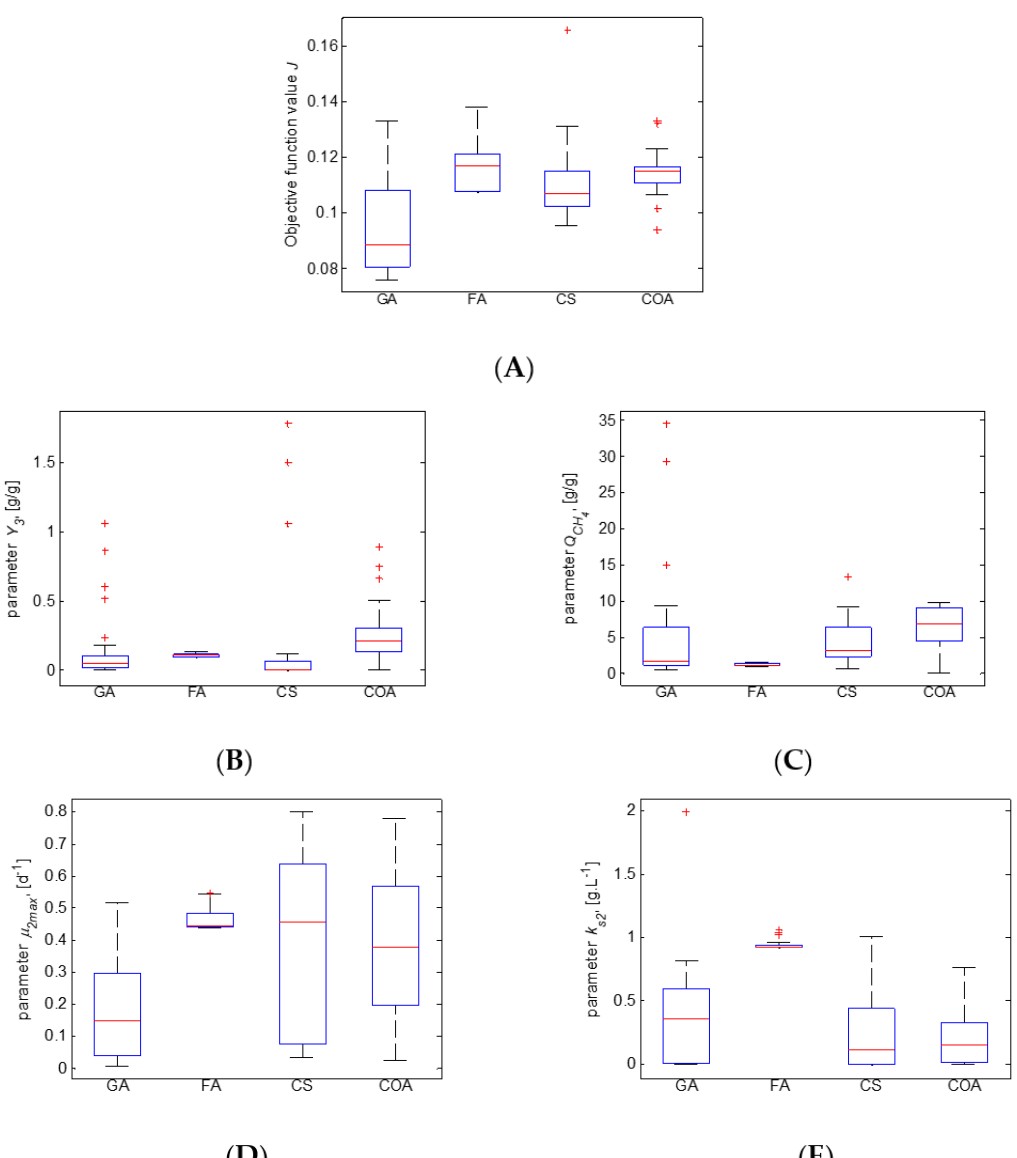

**Figure 3.** Box plot for the results of the second identification step. (**A**) objective function $J_2$; (**B**) parameter $Y_3$; (**C**) parameter $Y_{CH_4}$; (**D**) parameter $\mu_{2max}$; (**E**) parameter $K_{S_2}$.

As shown in Figure 3A, the median of the objective function values obtained from 30 runs of GA is better than that of the rest of the results, including the minimum achieved by the other algorithms. On the other hand, the SD in the case of GA is larger. The best SD is observed for the results of COA, followed by those of FA. The median of 30 runs of CS is better than the median of COA and FA. These observations are a reason to conclude that the advantages and disadvantages of the considered algorithms should be analyzed in depth. Subsequently, a hybrid metaheuristic can be designed to combine the advantages.

It is shown that the distribution of FA estimates of the model parameters is smaller than that of the rest of the algorithms. The FA SD is times smaller than the SD of GA, CS, and COA. The model parameters $\mu_{2max}$ and $K_{S_2}$ are more difficult to evaluate than the model parameters $Y_3$ and $Y_{CH_4}$. This is an interesting result, taking into account that the parameters $\mu_{2max}$ and $K_{S_2}$ are usually the most sensitive ones [60]. It is worth investigating the sensitivity of the parameters of the new mathematical model proposed here. Moreover,

outliers have been observed among parameter estimates, especially for the model parameter $Y_3$. Such outliers may appear in a sample of a normally distributed population, but this kind of result would be of interest to future research.

Friedman's non-parametric test [61] was used to determine whether or not there was a statistically significant difference between the GA, FA, CS, and COA-derived objective function values for the 30 runs. The test is performed in Matlab R2013a using the function *friedman*. The obtained results are shown in Figure 4.

**Friedman's ANOVA Table**

| Source | SS | df | MS | Chi-sq | Prob>Chi-sq |
|--------|------|-----|---------|--------|-------------|
| Columns | 2543.5 | 29 | 87.7069 | 32.99 | 0.0278 |
| Error | 6399 | 87 | 73.5517 | | |
| Total | 8942.5 | 119 | | | |

**Figure 4.** Friedman's test results.

Since the *p*-value is less than 0.05, the null hypothesis that the mean of the objective function is the same for all four algorithms can be rejected. This is sufficient evidence to conclude that the results obtained by GA, FA, CS, and COA are statistically significant.

For practical applications, it is important to use a method that is theoretically guaranteed to converge, robust method, a method that has such a guarantee starting from any initial solution [62]. The estimates presented from 30 runs of each metaheuristic and the statistical results discussed confirm the robustness of the GA, FA, CS, and COA considered here. The observed SD, especially for FA and COA, was 0.0094 and 0.0074, respectively. It is shown that the studied algorithms, starting from 30 different initial solutions, converge to very close (statistically equal) values of the objective function.

In summary, based on the results obtained and the analysis carried out, the model estimated by applying FA is assumed to be a better mathematical model of the TSAD process considered here. The obtained model parameters' estimates $p_1 = [\mu_{1max} = 0.017 \ \text{day}^{-1} \ K_{S_1} = 1.1 \ \text{g/L} \ Y_1 = 0.22 \ Y_2 = 10.28 \ Y_{H_2} = 8.12]$ and $p_2 = [\mu_{2max} = 0.44 \ \text{day}^{-1} \ K_{S_2} = 0.92 \ \text{g/L} \ Y_3 = 0.1 \ Y_{CH_4} = 0.99]$ are physically correct and adequate based on the published mathematical models of AD processes close to the one proposed here [54,55].

The proposed mathematical model of the TSAD process obtained by FA has satisfactory accuracy. It could be further used for process investigation as well as for process control and optimization.

## 4. Conclusions and Future Works

In this paper, a new mathematical model of two-stage anaerobic digestion of corn steep liquor has been proposed. The model has been presented as a set of five ordinary differential equations and two algebraic equations, representing seven dependent state variables (substrate concentration ($S_1$) in $BR_1$; microbial biomass concentration in $BR_1$ ($X_1$) and $BR_2$ ($X_2$); acetate formation in $BR_1$ ($Ac_1$) and $BR_2$ ($Ac_2$); flow rate of the hydrogen ($Q_{H_2}$) in the gas phase of $BR_1$ and methane flow rate $Q_{CH_4}$ in $BR_2$). The mathematical model consists of nine unknown model parameters divided into two groups, $p_1 = [\mu_{1max} \ K_{S_1} \ Y_1 \ Y_2 \ Y_{H_2}]$ and $p_2 = [\mu_{2max} \ K_{S_2} \ Y_3 \ Y_{CH_4}]$. Different metaheuristic algorithms, some of the popular algorithms of the swarm intelligence domain, have been applied to estimate the unknown parameters based on experimental data.

The feasibility of the Genetic algorithm, Firefly algorithm, Cuckoo search algorithm, and Coyote Optimization Algorithm applied to a parameter identification problem has been highlighted. The chosen metaheuristic algorithms have been adapted and implemented here for the first time for parameter estimation of a newly proposed nonlinear mathematical model of TSAD of corn steep liquor. To confirm the superiority of some of the algorithms, a comparison between the techniques has been made—using the observed numerical results, graphical results, and statistical analysis.

The simulation results of the parameters' identification of the proposed mathematical models show that GA, FA, CS, and COA techniques can predict the experimental results. The four metaheuristics have achieved similar results in modelling the process dynamics in $BR_1$. In the case of modelling the process dynamics in $BR_2$, the lowest value of the objective function has been achieved by GA ($J = 0.0761$). A better description of the process dynamics trend, however, has been obtained by FA ($J = 0.1075$). The observed objective function values of CS and COA are $J = 0.0913$ and $J = 0.0940$, respectively. It has been demonstrated that the considered metaheuristics are efficient and are also powerful algorithms for parameter identification of complex nonlinear models. As a result, a mathematical model of TSAD of corn steep liquor with a high degree of accuracy is proposed. The model can be used for a simulation of the process behaviour to gather enough information for planning subsequent laboratory experiments. The mathematical model can also be used to optimize the process based on a designed control system. A good model in control loop synthesis is required to obtain optimal control.

In addition, the numerical results have been compared based on box plots—a standard technique for analysis of the distribution of the obtained model estimates. The statistical results show that the parameter estimates $Y_3$ and $Y_{CH_4}$ are more easily obtained compared to the parameter estimates $\mu_{2max}$ and $K_{S_2}$. Such results are interesting, and it is worth conducting further research on the sensitivity of the model parameters to optimize the identification process.

Although some good results have been achieved with the application of the considered metaheuristics, there are still some limitations that need to be improved in future. First, the presence of several sets of raw experimental data is important for model validation. This is crucial, especially in the case of modelling non-linear, time-varying processes with interdependent variables, such as the one considered in this paper. Additional laboratory experiments are planned to be carried out. Second, it is well known that to achieve better performance, the input parameters of metaheuristic algorithms should be adjusted according to the problem domain. A joint set-up procedure [29] will be applied to the algorithms' parameters to significantly improve their performance. Finally, the results of the statistical analysis will be used for improving the efficiency of the algorithms. More numerical experiments will be conducted to identify the strengths and weaknesses of the studied metaheuristic algorithms. More powerful hybrid algorithms will be developed with a rapid convergence speed and robustness, as presented in [63,64].

**Author Contributions:** Conceptualization, O.R. and E.C.; methodology, O.R. and E.C.; software, O.R.; validation, O.R. and E.C.; formal analysis, O.R. and E.C.; investigation, O.R. and E.C.; data curation, O.R. and E.C.; writing—original draft preparation, O.R. and E.C.; writing—O.R. and E.C.; visualization, O.R. and E.C.; supervision, O.R.; project administration, E.C.; funding acquisition, E.C. All authors have read and agreed to the published version of the manuscript.

**Funding:** This research received no external funding.

**Institutional Review Board Statement:** Not applicable.

**Informed Consent Statement:** Not applicable.

**Data Availability Statement:** Not applicable.

**Acknowledgments:** This research has been supported by the Bulgarian National Science Fund, Grant No КП-06-Н46/4 "Experimental studies, modelling and optimal technologies for biodegradation of agricultural waste with hydrogen and methane production".

**Conflicts of Interest:** The authors declare no conflict of interest.

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
