# Peer review of "Metaheuristic Algorithms to Optimal Parameters Estimation of a Model of Two-Stage Anaerobic Digestion of Corn Steep Liquor"

_applsci, doi:10.3390/app13010199_

Round 1

Reviewer 1 Report

my comments are attached.

Author Response

Dear reviewer,

Thank you for your valuable comments and suggestions. Taking into account all of them, we believe that the quality of the paper has been improved significantly.

All improvements are presented in red colour in order to be easily identifiable by the editors and reviewers.

Attached are the detailed answers, point by point, to each of your comments.

Reviewer 2 Report

1. In Abstract: Anaerobic digestion (AD), please change to Anaerobic                                     Digestion (AD)

                     two-stage anaerobic digestion (TSAD), please change to Two-                          Stage Anaerobic Digestion (TSAD)

2. I cannot find any novelties in Abstract section. The abstract does not reflect the contribution of the study. Please, motivate more the abstract, trying to be more concise. Why this work is necessary?

3. In Introduction:  Anaerobic digestion (AD), please change to Anaerobic                                     Digestion (AD)

                               two-stage anaerobic digestion (TSAD), please change to                                 Two-Stage Anaerobic Digestion (TSAD)

                               Corn steep liquor (CSL), please change to Corn Steep                                       Liquor (CSL)

4. In Introduction: Authors can present better novelty of their work

5.In section 2: ordinary differential equations (ODEs) change to Ordinary                                Differential Equations (ODEs)

6. Why authors use of these methods ''Genetic Algorithm (GA), Firefly Algo rithm (FA), Cuckoo Search Algorithm (CS) and Coyote Optimization Algorithm (COA)''???They should explain the reason of use of these algorithms.

7. The literature review is brief. Some of the included papers could be briefly described. Also, a general overview of the topic (metaheuristic algorithms) could also be included. For instance, the following could be added:

*(2022). A sustainable-resilience healthcare network for handling COVID-19 pandemic. Annals of operations research, 312(2), 761-825.

*(2021, August). Designing a New Supply Chain Network Considering Transportation Delays Using Meta-heuristics. In International Conference on Intelligent and Fuzzy Systems (pp. 570-579). Springer, Cham.

8. The writing of the paper needs a lot of improvement in terms of grammar, spelling, and presentations. The paper needs careful English polishing since there are many typos and poorly written sentences.

Some examples are as the following:

*     Check the usage of the commas carefully.

*     Check the articles including "a", "an" and "the".

*     Check the required and unneeded blank spaces.

9.What method did use for validation model? I cannot find it.

10.Authors can change the title section 5 to Conclusion and future works.

11.No managerial and theoretical implications of the findings is narrated in the paper.

12.Limitations of the model and solution approach are not discussed.

Author Response

Dear reviewer,

Thank you for your valuable comments and suggestions. Taking into account all of them, we believe that the quality of the paper has been improved significantly.

All improvements are presented in red colour in order to be easily identifiable by the editors and reviewers.

Following are the detailed answers, point by point, to each of your comments.

Author Response

(The authors gave the same response as above.)

Round 2

Reviewer 1 Report

Author explanations are satisfactory. The current version is acceptable.

Reviewer 2 Report

Accepted

Reviewer 3 Report

The paper is revised according to suggestions and concerns. The paper's organization and presentation seems to have improved as a result of the revision in a logical and sensible way. Problem statement, the methods and the obtained results are better presented. No more comments for changes from this reviewer’ side and the paper in current situation seems acceptable for publication.